# Label Uncertainty Suppression Based on Heterogeneous Siamese Network for Neonatal Pain Assessment in Uncontrolled Conditions

Songyang Lin
*College of Electrical Engineering*
*Zhejiang University*
Hangzhou, China
linsongyang@zju.edu.cn

Huaiyu Zhu
*College of Information Science*
*and Electronic Engineering*
*Zhejiang University*
Hangzhou, China
zhuhuaiyu@zju.edu.cn

Kai Tong
*College of Information Science*
*and Electronic Engineering*
*Zhejiang University*
Hangzhou, China
tongkai@zju.edu.cn

Yisheng Zhao
*Hangzhou Institute of*
*Advanced Study*
*University of Chinese Academy of*
*Sciences*
Hangzhou, China
zhaoyisheng@ucas.ac.cn

Shuohui Chen
*Children's Hospital*
*Zhejiang University School of*
*Medicine*
Hangzhou, China
chcsh2@zju.edu.cn

Yun Pan
*College of Information Science*
*and Electronic Engineering*
*Zhejiang University*
Hangzhou, China
panyun@zju.edu.cn

*Abstract*—Reliable assessment of neonatal pain is essential for timely clinical intervention. However, the ambiguity and variability of facial expressions in real-world clinical environments pose significant challenges. Although deep learning-based approaches have shown promise in automated neonatal pain assessment (NPA), their performance is often hindered by label noise, annotation ambiguity, and limited labeled data. To address these issues, we propose an uncertainty-aware dual-branch heterogeneous Siamese network that suppresses label noise and enhances the robustness of NPA. The framework incorporates an uncertainty-guided rank regularization module and a pseudo-label correction strategy, enabling dynamic label refinement during training. Moreover, visual analysis based on Gradient-weighted Class Activation Mapping (Grad-CAM) demonstrates that the model learns pain-intensity-sensitive attention patterns, providing interpretability and practical guidance for clinical annotation, particularly under occlusion conditions. Extensive experiments on our neonatal pain dataset show that the proposed method achieves superior accuracy compared to existing state-of-the-art label correction approaches, indicating the effectiveness and reliability of our model for automated NPA in challenging real-world settings.

*Index Terms*—Neonate pain, label uncertainty, facial expression, heterogeneous Siamese network, uncontrolled conditions.

## I. INTRODUCTION

The assessment of procedural pain — defined as the evaluation of pain experienced by neonates during routine medical procedures such as heel lancing, venipuncture, or vaccination — is the most common and representative application scenario

This work was supported in part by the National Natural Science Foundation of China (62306272), in part by the Pioneer and "Leading Goose" Research and Development Program of Zhejiang (2024C03027), and in part by the Research Project of Chinese Nursing Association (ZHKY202312). (*Corresponding author: Huaiyu Zhu; Yun Pan.*)

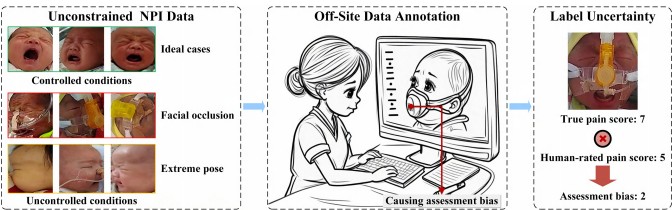

Fig. 1. Label bias caused by real-world interferences

for neonatal pain management in clinical practice [1]. Such assessments typically rely on the proficiency of medical staff in utilizing various standardized pain scales, alongside their extensive prior knowledge and clinical experience. Accurate assessment must be conducted manually by trained professionals using validated scales, which is labor-intensive, time-consuming, and highly subjective [2].

In recent years, artificial intelligence technologies for image analysis have demonstrated initial success in neonatal pain assessment (NPA). Facial expression is the most important factor in assessing neonatal pain and can be obtained through neonatal pain images (NPIs) [3]. However, substantial challenges remain in real-world clinical settings, where NPIs might be captured in uncontrolled conditions. Although researchers have preliminary explored methods for analyzing noisy neonatal pain expressions in real-world neonatal ward scenarios by Generative Adversarial Network (GAN)-based facial representation learning [4], existing approaches generally suffer from limited generalization capability and poor robustness when facing unconstrained NPI data [5-9].

In real-world NPIs, neonatal facial appearance can be easily

altered by occlusion and pose variation (Fig. 1). These interferences affect the ability of professional medical staff to accurately judge neonatal pain levels during off-site data annotation for large-scale dataset construction [10]. As a result, pain labels may contain uncertainty, which can mislead the training of NPA models. Hence, it is of great significance to suppress the uncertainty of unconstrained NPI labels during the training process, thereby eliminating potential errors introduced by the data for a robust NPA method in uncontrolled conditions.

The development of label correction techniques focused on label completion and error rectification. Okamura et al. [11] proposed a label correction method based on the memorization effect of neural networks, in which early-stage network predictions are utilized to dynamically replace noisy labels, thus avoiding reliance on prior knowledge of noise rates. In the context of facial expression recognition under label noise, Zhang et al. [12] introduced a unified architecture that enhances label robustness via attention erasure and consistency learning, effectively suppressing unreliable attention regions. Wang et al. [13] proposed the Self-Cure Network (SCN), which integrates uncertainty weighting, rank regularization, and conditional relabeling to address ambiguous annotations. Furthermore, Yi et al. [14] explored end-to-end modeling of label confidence and incorporated graph-based reasoning to estimate label quality, further improving the generality and robustness of label correction strategies.

Recent efforts have shifted from simple sample selection to structured and task-specific label correction mechanisms based on multi-source reasoning. Co-teaching and uncertainty-aware pseudo-label correction have been integrated as popular and effective strategies for learning with noisy labels. Co-teaching, as a robust learning framework, involves training two networks simultaneously with the mutual selection of low-loss samples to guide each other, thereby preventing overfitting to noisy labels [15]. This method has been widely applied in image classification, medical diagnosis, and industrial defect detection, with various extensions proposed to improve its practicality and robustness.

In the field of label correction with Siamese networks, Wang et al. [16] utilized two interactive LEAStereo networks in an unsupervised stereo matching task to mutually learn occlusion region information, effectively improving recognition accuracy. Chen et al. [17] developed the dual clustering co-teaching approach for unsupervised person re-identification tasks, generating complementary pseudo-labels through two networks employing different clustering parameters, thus successfully mitigating noise interference in the pseudo-labels. Although the aforementioned studies have demonstrated promising performance using co-teaching frameworks, they generally rely on homogeneous dual-network architectures. Such homogeneity can lead to insufficient diversity between the two networks, potentially limiting their effectiveness in distinguishing clean samples from noisy ones.

To address this issue, we propose a heterogeneous Siamese network that inherently encourages diversity between different branches. Specifically, we design a dual-branch uncertainty suppression network comprising ResNet-50 [18] and EfficientNet-V2 [19] models. This design aims to identify and suppress interference from uncertain NPI labels and samples near decision boundaries from different learning perspectives. Additionally, to enhance the discrimination capability of the uncertainty weighting module and ensure reliable sample identification, we incorporate a hierarchical regularization term into the loss function as an auxiliary supervision signal. This facilitates dynamic optimization of the model training process and enables more accurate label correction and discrimination capabilities. The proposed method could overcome common weaknesses observed in prior homogeneous co-teaching frameworks and achieve a more reliable and effective smart NPA method for real-world scenarios. The main contributions of our paper are as follows:

- We propose a label uncertainty suppression mechanism based on a novel Heterogeneous Siamese Network.
- We introduce a hierarchical regularization term to enhance the sample discrimination capability of the uncertainty weighting module.
- The experimental results demonstrated that the proposed method achieves enhanced robustness and performance for neonatal pain assessment in unconstrained scenarios.

## II. METHODS

### A. Heterogeneous Siamese Network

Samples with label uncertainty can cause the NPA model to deviate from the true distribution of the training dataset, often resulting in overfitting to non-global features and increasing the model's generalization error. Conventional single-branch networks are particularly prone to either overfitting noise or underfitting effective features when facing high label uncertainty. To enhance discriminative performance, we propose a dual-branch heterogeneous Siamese network architecture that performs collaborative generalization error optimization across both data and algorithmic domains by leveraging complementary, non-weight-sharing network structures, as shown in Fig. 2. We further integrate an uncertainty-aware discrimination mechanism with hierarchical regularization into the loss function. This mechanism leverages reliability-weighted pseudo-labeling to dynamically high-light reliable annotations while suppressing the influence of uncertain labels.

### B. Uncertainty Discrimination

A key challenge in suppressing label uncertainty lies in how to obtain reliable label uncertainty analysis results and leverage these results during training to emphasize reliable annotations while mitigating the influence of potentially uncertain samples. In this work, we propose a dual-branch heterogeneous Siamese network as the backbone for pain information fusion and analysis. This architecture is constructed using two neural networks with distinct structural configurations. This design endows the Siamese network with diversified perceptual learning capabilities, enabling it to capture label uncertainty from multiple perspectives.

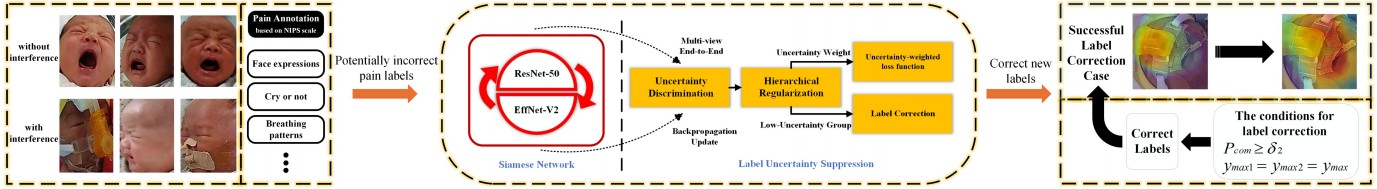

Fig. 2. Diagram of label correction principle

Building on this multi-view Siamese framework, we adopt a mutually supervised uncertainty discrimination strategy. Specifically, we design an uncertainty discrimination module that assigns an uncertainty weight to each sample. These weights are regulated by a rank regularization module to ensure that the learned uncertainty weights are meaningful. This module is jointly optimized with the Siamese network in an end-to-end fashion, not only providing guidance for training via sample reweighting but also receiving optimization signals from distinct network perspectives.

The uncertainty discrimination module consists of a linear fully connected (FC) layer followed by a Sigmoid activation function. The calculation of the uncertainty weight is defined in Equation (1), where $\alpha_i$ denotes the uncertainty weight of the i-th sample $x_i$, $W_\alpha$ represents the parameters of the FC layer, and $\sigma(\cdot)$ denotes the Sigmoid function.

$$\alpha_i = \sigma(W_a^T x_i) \tag{1}$$

### C. Hierarchical Regularization

The rank regularization module constrains the attention weights by first fusing the uncertainty weights $\alpha_1$ and $\alpha_2$ predicted by the two branches of the heterogeneous Siamese network. The fusion is performed using a multiplicative strategy ($\alpha_1 \times \alpha_2$) to obtain a joint uncertainty measure for each sample. Based on a predefined ratio $\beta$, the fused attention weights are then partitioned into a high-uncertainty group and a low-uncertainty group. To impose a ranking structure, an explicit margin is enforced between the mean uncertainty weights of the two groups, thereby encouraging the network to distinguish between reliable and unreliable samples. This constraint guides the uncertainty estimation module to learn meaningful weights that promote confident annotations while suppressing ambiguous ones. The regularization is implemented via the rank-regularization loss functions defined in Equations (2) and (3), where $\delta_1$ is a hyperparameter, and $\alpha_H$ and $\alpha_L$ represent the average uncertainty weights of the high-uncertainty group (size $M$) and the low-uncertainty group (size $N-M$), respectively.

$$\mathcal{L}_{RR} = \max\{0, \delta_1 - (\alpha_H - \alpha_L)\} \tag{2}$$

$$\alpha_H = \frac{1}{M}\sum_{i=0}^{M}\alpha_i, \quad \alpha_L = \frac{1}{N-M}\sum_{i=M}^{N}\alpha_i \tag{3}$$

### D. Label Rectification Strategy

Upon obtaining reliable uncertainty weights, we design a dual approach to mitigate the variance and bias components of generalization error introduced by label uncertainty. Specifically, we propose an uncertainty-weighted loss function and a mutual-supervision-based uncertainty-driven label rectification scheme.

To effectively leverage uncertainty weights for guiding the training of intelligent analysis models, we adopt a logarithmic weighting scheme that modifies the conventional multi-class cross-entropy loss, as shown in Equation (4). In this formulation, $y_i$ denotes the ground-truth label of sample $x_i$, and $W_j$ represents the classifier parameters corresponding to the j-th class among the total $C$ categories.

$$\mathcal{L}_{WCE} = -\frac{1}{N}\sum_{i=1}^{N}\log\frac{e^{\alpha_i \mathbf{W}_{y_i}^\top x_i}}{\sum_{j=1}^{c}e^{\alpha_i \mathbf{W}_j^\top x_i}} \tag{4}$$

For samples categorized within the low-uncertainty group, we introduce an uncertainty-aware label rectification mechanism. If both branches of the heterogeneous Siamese network yield consistent predicted labels for a given sample, and the difference between the Softmax prediction probabilities for the most confident class and the originally assigned label exceeds a specified threshold, the sample is reassigned a pseudo-label. This process is governed by a margin-based thresholding condition, as defined in Equation (5), where $y'$ denotes the rectified label, $y_{org}$ and $y_{max}$ refer to the original label and the label associated with the maximum predicted probability, respectively.

$$y' = \begin{cases} y_{\max}, & \text{if condition} \\ y_{\text{org}}, & \text{otherwise} \end{cases} \tag{5}$$

To enable label correction under high-confidence agreement, this strategy leverages mutual supervision between two structurally distinct branches and applies appropriately defined thresholds. The fused maximum prediction probability, denoted as $P_{com}$ is computed as the average of their individual maximum prediction probabilities.

$$P_{\text{com}} = \frac{P_{\max 1} + P_{\max 2}}{2} \tag{6}$$

where $P_{max1}$ and $P_{max2}$ indicate the maximum prediction probabilities from the first and second branches, labels are corrected only when both branches predict the same class and the combined confidence exceeds a predefined threshold. The

conditions can be formally expressed by Equation (7), where $y_{max1}$ and $y_{max2}$ denote the predicted labels from the two branches of the Siamese network and $\delta_2$ and represent the predefined thresholds.

$$P_{\text{com}} \geq \delta_2$$
$$y_{\text{max1}} = y_{\text{max2}} = y_{\text{max}} \qquad (7)$$

Notably, these rectified samples may subsequently be assigned higher uncertainty weights in future training iterations, enabling the model to adaptively correct uncertain annotations in a self-supervised manner throughout the training process.

Specifically, an exponential moving average (EMA) mechanism is further employed to perform gradient-free parameter tracking of the dual-branch Siamese network. This strategy enables smooth tracking of the parameter states of the heterogeneous branches throughout training, allowing more stable predictions to be utilized for pseudo-label correction of low-uncertainty samples in the later training stages. The EMA updates the tracked parameters by computing a weighted average between the historical parameters and the current ones at each iteration. The update rule is defined in Equation (8).

$$\theta_t^{\text{EMA}} = \lambda \cdot \theta_{t-1}^{\text{EMA}} + (1 - \lambda) \cdot \theta_t \qquad (8)$$

Here, $\theta^{EMA}$ denotes the current model parameters, $\theta_t$ represents the EMA-tracked parameters, and $\lambda$ is the smoothing coefficient and is set to 0.999 based on experience.

The proposed mutual-supervision-based uncertainty suppression network is designed to address the challenges posed by label noise and boundary samples, which often compromise model stability and generalization. This framework adopts a collaborative training strategy composed of two heterogeneous branches with non-shared parameters. Specifically, a dual-network architecture based on ResNet-50 [18] and EfficientNet-V2 [19] is constructed to extract pain-related feature representations from structurally diverse perspectives. For each sample, an uncertainty weight is predicted by a lightweight fully connected (FC) network to quantify the reliability of its label. Subsequently, based on the ranked uncertainty weights, a rank regularization term is introduced to enforce a margin between the mean weights of high- and low-uncertainty groups. This encourages the network to clearly distinguish between reliable and unreliable samples, thereby enhancing its robustness to noisy labels. Label correction is performed in a controlled manner by integrating three criteria: prediction consistency across both branches, geometric mean confidence, and the confidence margin between the original and predicted labels within each branch. This correction mechanism improves label quality during training and promotes more stable model convergence.

## III. RESULTS

### A. Dataset Preparation and Processing

This study was approved by the Ethics Committee of the Children's Hospital Zhejiang University School of Medicine

TABLE I
DETAILS FOR THE DATASET

| Subjects | 613 | | | |
|---|---|---|---|---|
| Gender | Female: 252, Male: 361 | | | |
| Gestational time | 1d–910d (avg: 55.78d) | | | |
| Pain levels | Neonatal Infant Pain Scale (NIPS) 0: 76, 1: 25, 2: 12, 3: 20, 4: 21, 5: 32, 6: 177, 7: 250 | | | |
| Clinical procedures | Arterial blood collection | 80 | Finger tip blood | 85 |
| | Heel prick | 61 | Intravenous injection | 46 |
| | Tape removal | 68 | Sputum suction | 65 |
| | Venipuncture catheterization | 75 | Retention enema | 73 |
| | Wound dressing change | 31 | Others | 29 |

(Approval No. 2023-IRB-0217-P-01). Guardians of the participants fully understood the data collection procedure and approved the recording of the data for our research use with written consent forms. Following this approval, key frames corresponding to peak pain moments were extracted from 613 videos capturing unconstrained neonatal pain-inducing procedures. Each frame was independently scored by experienced hospital nurses using the Neonatal Infant Pain Scale (NIPS) [20] and the assigned scores were used as the reference labels for each video. Facial regions were first detected and cropped from the selected key frames using the neonatal face detection system [21]. The extracted facial images were then subjected to a standard preprocessing pipeline: each image was first converted from OpenCV format to PIL, resized to ensure the shorter side is 256 pixels while maintaining aspect ratio, center-cropped to 224×224, converted to a PyTorch tensor, and finally normalized using ImageNet statistics (mean = [0.485, 0.456, 0.406], std = [0.229, 0.224, 0.225]). This normalization aligns the input distribution with that of most ImageNet-pretrained models. The dataset was split into training and validation sets at a ratio of 70% and 30%. Detailed dataset statistics are provided in Table I.

The dataset was partitioned according to the annotated pain levels (ranging from 0 to 7), with 426 samples allocated for training and 187 for testing. The label distribution was maintained to be approximately uniform across both subsets. To simulate real-world annotation disturbances, label noise was artificially introduced into 10% of the training samples for each class. Specifically, noisy labels were generated by randomly adding or subtracting a bias from the original label, as formulated in Equation (9), where $y_{org}$ denotes the original label, $\hat{y}_{org}$ represents the label after noise injection, and $\epsilon$ draws a random integer from the discrete uniform distribution over the set $\{1, 2, 3\}$. We further constrained the noisy labels to ensure they were no less than 0 and no greater than 7.

$$\hat{y}_{\text{org}} = y_{\text{org}} \pm \epsilon \qquad (9)$$

### B. Implementation details

The proposed heterogeneous Siamese network is developed in Python 3.7, Pytorch 2.7.0. The training and evaluation were conducted on a single NVIDIA RTX 8000 GPU. The AdamW optimizer was used with an initial learning rate of 1e-4 and a weight decay of 1e-4. A cosine annealing scheduler was

applied with a total cycle of 70 epochs. The batch size was set to 64. The proposed heterogeneous Siamese Network was built upon pretrained ResNet-50 [18] and EfficientNet-V2 [19] backbones, with an output dimension of 8 to support multi-class pain intensity classification. A dropout rate of 0.1 was adopted to reduce overfitting.

The loss function combined cross-entropy loss and rank regularization, with the uncertainty grouping parameter ($\beta$) set to 0.5. The value of the rank regularization $\delta_1$ is selected via grid search within the range [0.02, 0.2] with a step size of 0.02, and the optimal value is determined to be $\delta_1 = 0.1$. Dynamic label correction is initiated from the 10th training epoch, leveraging a collaborative prediction mechanism based on the dual-model framework. To further enhance the flexibility and adaptability of threshold selection, we adopt a dynamic adjustment strategy for $\delta_2$. In the early stage of training (epoch $\leq 50$), a grid search is conducted over the interval [0.3, 0.5] with a step size of 0.05, resulting in the selection of a fixed optimal threshold $\delta_2 = 0.4$. In the later stage ($50 < $ epoch $\leq 70$), this threshold is gradually increased in a linear manner up to $\delta_2 = 0.8$, which is also determined as the optimal final value based on grid search within the interval [0.6, 0.9], using the same step size of 0.05.

### C. Effectiveness of Label Correction with EMA

To evaluate the role of the Exponential Moving Average (EMA) mechanism in the label correction strategy, we conducted a controlled ablation study by removing the EMA update module while keeping all other training settings identical. To ensure robustness, we performed five independent runs of the ablation experiment. Throughout training without EMA, no label corrections were observed—the number of corrected labels remained zero. This is because the two networks failed to reach prediction consistency, making it difficult to trigger relabeling events.

In contrast, when EMA is applied, label corrections begin to occur around epoch 40. At this stage, the predictions of the two networks begin to align. This consistency enables the system to confidently identify and relabel incorrect targets. The improved agreement is a result of the EMA mechanism helping the model converge more rapidly toward the optimal solution. As shown in Table II, the overall performance of the model degraded significantly without EMA. Specifically, the test accuracy dropped from 80.78% to 76.56%, reflecting an average decrease of 4.24%, which indicates a notable decline in the model's generalization capability.

These results demonstrate the effectiveness of the label correction strategy in mitigating the impact of noisy labels and enhancing model generalization. Further analysis reveals that the correction mechanism dynamically updates the labels of low-confidence samples through dual-branch agreement and confidence-guided relabeling, thereby reducing the negative effects of incorrect supervision during training.

As the relabeling trends were highly consistent across all runs, we present the results from one representative trial in Fig. 3 for detailed analysis. Fig. 3 shows the number of relabeled

TABLE II
EFFECT OF RELABELING WITH EMA ON ACCURACY

| Times | Without EMA (%) | With EMA (%) |
|---|---|---|
| 1 | 73.89 | 80.56 |
| 2 | 76.67 | 80.56 |
| 3 | 78.33 | 80.00 |
| 4 | 80.00 | 81.67 |
| 5 | 73.89 | 81.11 |
| **Average** | **76.56±2.70** | **80.78±0.64** |

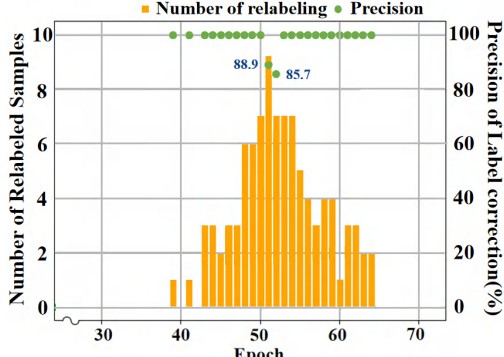

Fig. 3. Number and precision of labels corrected

samples across epochs and the precision of the label correction process. Initially, the relabeling frequency is relatively low; as the two branches become more consistent in later stages of training, the number of corrected labels increases, reaching a peak. Subsequently, as the model converges, the relabeling frequency declines gradually. This trend indicates that the model's ability to identify and correct mislabeled samples improves progressively and eventually stabilizes. In the majority of cases, the corrected labels achieved 100% accuracy, with only a few instances exhibiting anomalies. These results provide strong evidence for the reliability and correctness of the label correction method.

In summary, these observations demonstrate that effective relabeling is a downstream outcome of training stability and prediction alignment—both of which are fostered by the EMA technique.

### D. Effectiveness and Superiority of heterogeneous Siamese Network

To evaluate the effectiveness of the heterogeneous Siamese network architecture in representation learning, we compared the performance of three structural configurations: a single-branch ResNet, a single-branch EffNet, and the proposed dual-branch uncertainty-aware Siamese network. All models were trained under identical hyperparameter settings and data splits, and each configuration was independently repeated five times. The results are summarized in Table III.

As shown in the experimental outcomes, the heterogeneous Siamese network consistently outperformed both single-branch counterparts across all trials, demonstrating a significant improvement in test accuracy. Specifically, the ResNet branch

TABLE III
ACCURACY OF SIAMESE, SINGLE-BRANCH, AND OTHER NETWORKS

| Times | Res [18] (%) | Eff [19] (%) | Era-Atten[12] (%) | Self-Cure[13] (%) | Our Method (%) |
|---|---|---|---|---|---|
| 1 | 76.67 | 76.67 | 77.01 | 78.89 | 80.56 |
| 2 | 76.67 | 74.44 | 77.54 | 76.67 | 80.56 |
| 3 | 78.33 | 73.33 | 78.07 | 77.22 | 80.00 |
| 4 | 76.67 | 78.33 | 76.47 | 78.89 | 81.67 |
| 5 | 77.78 | 76.67 | 78.60 | 78.89 | 81.11 |
| Average | $77.22 \pm 0.78$ | $75.89 \pm 1.99$ | $77.54 \pm 0.8$ | $78.11 \pm 1.08$ | $80.78 \pm 0.64$ |

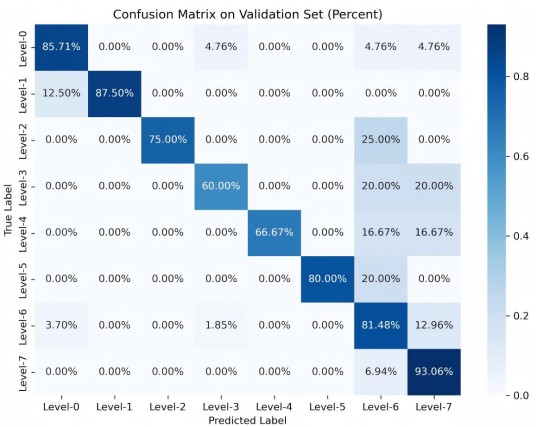

Fig. 4. The confusion matrix of the model's predictions

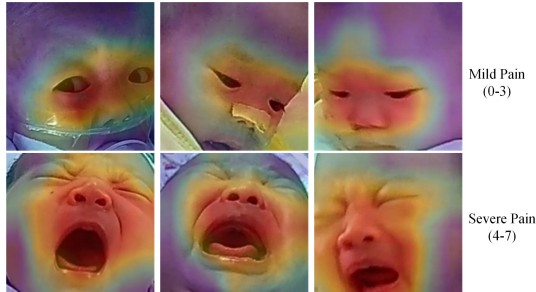

Fig. 5. Model attention on unobstructed neonatal faces

achieved an average accuracy of $(77.22 \pm 0.78)\%$, the EffNet branch reached $(75.89 \pm 1.99)\%$, while the heterogeneous Siamese network attained a superior performance of $(80.78 \pm 0.63)\%$. These results suggest that the two network branches are complementary in feature extraction and contribute jointly to enhanced discriminative capability.Fig. 4 presents the confusion matrix of the heterogeneous Siamese network from a representative experiment, further illustrating the model's discriminative performance across different classes.

We further compared our approach with other representative label correction methods. The results demonstrate that the heterogeneous Siamese network consistently outperformed both the Erasing-Attention [12] and Self-Cure models [13] across five independent trials, yielding a notable improvement in classification accuracy. Specifically, the Erasing-Attention model achieved an average accuracy of $(77.54 \pm 0.84)\%$, while the Self-Cure model reached $(78.11 \pm 1.08)\%$. In contrast, the heterogeneous Siamese network delivered superior performance, attaining an average accuracy of $(80.78 \pm 0.63)\%$, which represents an improvement of 3.24% and 2.67% over the previous two methods, respectively.

*E. Indicating potential bias of label*

To quantitatively evaluate the correction capability of our method for incorrect labels, we previously injected 10% noise into our carefully annotated dataset. However, even without adding the 10% noise (considering that perfectly accurate annotations are merely theoretical), our model still performed label modifications. To investigate what types of label modifications occurred in our model when applied to strictly annotated labels and what these modifications imply,

We employed Grad-CAM to visualize the model's attention during both pain assessment and label correction processes. Our analysis revealed that the label modifications occurring under these conditions might indicate potential biases within the strictly annotated labels themselves.

Concretely, on one hand, for neonates with unobstructed facial regions, Grad-CAM results indicate that the model primarily focuses on the eye region in cases of mild pain, while shifting attention to the nose and mouth in cases of severe pain. This phenomenon may be attributed to the hierarchical nature of pain expression in neonates. Specifically, mild pain often manifests through subtle facial cues such as eye tightening or slight brow contraction, whereas severe pain is characterized by more pronounced expressions like nasal flaring or mouth opening, which engage broader facial muscle groups. As illustrated in Fig. 5, the attention heatmaps clearly demonstrate this shift in focus: in mild pain conditions, highlighted regions concentrate around the eyes, while in severe pain conditions, the attention extends to the nose and mouth areas.

These findings suggest that, in future annotation processes, human annotators should broaden their focus to include the overall facial muscle activity, especially when differentiating between moderate and severe pain levels. Particular attention should be paid to movements around the nose and mouth, which are more indicative of high-intensity pain.

On the other hand, in the absence of label noise, the model's label correction behavior is predominantly observed in neonates whose facial regions are partially occluded by medical equipment, such as respiratory masks. These occlusions may lead to human annotation errors, resulting in potentially incorrect pain labels. The heterogeneous Siamese network demonstrates a strong capacity to correct such cases. Interestingly, even in these occluded instances, the model exhibits attention patterns consistent with pain intensity—focusing on the eyes during mild pain corrections and on the nose and mouth during corrections of severe pain. Fig. 6 presents Grad-CAM heatmaps of several representative cases where the model successfully corrected mislabeled samples. These visualizations clearly show the model's ability to locate meaningful facial cues despite partial occlusion and to align its attention with the underlying pain level.

The model's ability to successfully revise occluded samples, alongside its intensity-aware attention patterns, offers practical

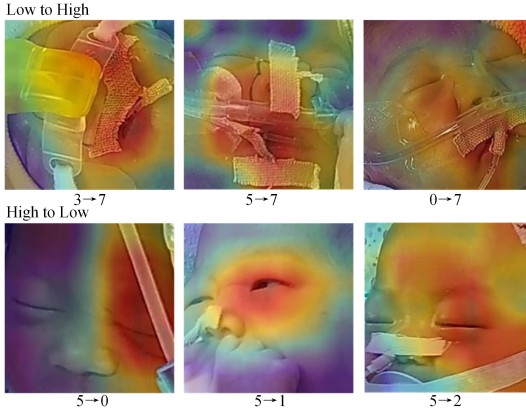

Low to High

3→7    5→7    0→7

High to Low

5→0    5→1    5→2

Fig. 6. Attention on obstructed faces and label correction

guidance for human annotation. Specifically, it highlights the need for labeling strategies that are sensitive to pain intensity and adaptive to facial occlusions, thereby contributing to improved label accuracy and overall dataset quality.

## IV. CONCLUSION

To reduce the impact of label noise on real-world NPA, we propose a dual-branch heterogeneous Siamese network to suppress the label uncertainty introduced by the off-site data annotation of NPIs. By incorporating rank regularization and a pseudo-label correction mechanism, the proposed approach enables dynamic refinement of noisy labels and enhances model robustness. Experimental results show that our framework achieves a classification accuracy of 80.78%, showing that our framework consistently outperforms single-branch networks in classification. Furthermore, ablation studies demonstrate the effectiveness of the pseudo-label correction with EMA mechanisms, which contribute additional accuracy gains of 4.24%, respectively. In comparison with existing state-of-the-art label correction methods, i.e., Erasing-Attention and Self-Cure, our approach achieves accuracy improvements of 3.24% and 2.67%, respectively, across multiple trials, indicating its superior capability in handling noisy and ambiguous labels.

In addition, Grad-CAM visualizations reveal that the model exhibits pain-intensity-sensitive attention patterns—focusing on the eyes for mild pain and on the nose and mouth for severe pain—and is capable of correcting potentially incorrect labels, particularly in cases of facial occlusion. This not only facilitates more reliable automated pain assessment but also provides annotation guidance for improving label accuracy in challenging clinical scenarios. In future work, we plan to further explore the application of this method in the analysis of pain behavior indicators other than facial expressions, e.g., arm and leg movements, and promote it to the analysis of neonatal pain videos.

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
