# OpenReview forum: "Label Uncertainty Suppression Based on Heterogeneous Siamese Network for Neonatal Pain Assessment in Uncontrolled Conditions"
_IEEE.org/EMBS/BHI/2025/Conference — BHI 2025_

### Official Review · Reviewer_Q5L6 · 2025-07-14
**Review of "Label Uncertainty Suppression Based on Heterogeneous Siamese Network for Neonatal Pain Assessment in Uncontrolled Conditions"**

**Confidence:** 4
**Clarity Of Writing:** great
**Clinical Significance:** good
**Methodological Novelty:** great
**Overall Rating:** 7
**Final Rating:** 8

**Experiments And Results:**

great

**Questions For The Authors:**

•  How does the performance of the proposed model compare to that of clinicians in assessing neonatal pain from facial expressions? Including such a comparison would help clarify the model’s potential clinical utility and practical relevance.
•  To what extent is the model generalisable across neonates from diverse ethnic backgrounds? Given possible variability in facial features, it would be valuable to assess or discuss the model’s robustness across demographic groups.

**Strengths:**

The paper addresses a clinically relevant problem. It proposes a technically sound solution to the challenge of label uncertainty in AI-based medical image classification. The method is clearly explained, with sufficient implementation details to support reproducibility by IEEE BHI readers. The proposed framework has potential translational impact both within neonatal pain assessment and across broader domains of clinical informatics where dealing with weak or noisy labels is common.

**Summary Of The Paper:**

The paper opens by highlighting challenges in the clinical management of neonatal pain, particularly the lack of objective and practical assessment solutions. It introduces artificial intelligence (AI) as a promising avenue for estimating neonatal pain based on facial expressions extracted from images. However, it notes that current AI approaches struggle to generalise across real-world conditions, where facial occlusions, pose variations, and other artefacts are common. As a result, data sets are difficult to build, and the labelling process is uncertain. This uncertainty in ground truths may mislead model training.  The paper critiques existing methods for addressing label uncertainty, including label completion, error rectification, and multi-source reasoning, arguing that these approaches remain insufficient.
To overcome these limitations, the paper proposes a novel dual-branch heterogeneous Siamese Network designed to suppress label uncertainty.  This architecture integrates an uncertainty discrimination module that weights each sample based on its reliability, alongside hierarchical regularisation and label rectification strategies. These components guide the model to learn preferentially from more reliable samples while minimising the influence of uncertain ones. The model is trained and validated on a dataset of 613 neonates, annotated by a nurse.
Two main findings are reported:  The proposed uncertainty suppression strategy improves classification accuracy by approximately 4%. The dual-branch heterogeneous Siamese Network outperforms both single-branch alternatives and other models, achieving an accuracy of 80.78% compared to less than 78.11% for competing approaches. The model also demonstrates robustness to noise.

**Weaknesses:**

•  In the abstract, the clarity of the abbreviation “Grad-CAM” would be improved by first writing out “Gradient-weighted Class Activation Mapping” in full before introducing the acronym.
•  In the introduction, the opening sentence would benefit from a more globally representative reference, as the cited study focuses on Ethiopia. In case the statement refers specifically to clinical practice in Ethiopia, it should be clarified.
•  In the introduction, the term “prototypical” in the first sentence is imprecise and may confuse readers. A clearer or more specific alternative would enhance clarity.
•  In the introduction, briefly defining the term “procedural pain” would support accessibility for IEEE-BHI readers who are not clinical experts.
•  In the introduction, the statement regarding the challenges of capturing NPIs in uncontrolled clinical environments would be strengthened by the inclusion of a supporting reference.
•  In the introduction, spelling out “Generative Adversarial Network” before introducing the abbreviation “GAN” would improve clarity, particularly for interdisciplinary readers.
•  In the introduction, the sentence beginning with “As neonatal facial appearance in real-world NPIs…” is overly long and would benefit from being split into shorter statements to enhance readability.
•  In Figure 1, adherence to IEEE BHI formatting requirements regarding figure placement (i.e. top or bottom of a column) should be ensured.
•  In the introduction, the claim that a given method has been widely applied in multiple domains would be better supported by a specific reference.
•  In the introduction, paragraphs 4 to 6 present existing approaches (e.g. label completion, multi-source reasoning) primarily in terms of their success. This structure may give the impression that the research problem is already resolved. A more balanced argument that also highlights the limitations of these methods would better motivate the need for the proposed solution.
•  In the introduction, following the review of related work, the advantages of the proposed heterogeneous Siamese network should be made more explicit in relation to the limitations of previous methods. This would strengthen the articulation of the paper’s novelty and relevance.
•  In the introduction, the phrase “In this field” at the start of paragraph 6 would benefit from clearer specification (e.g. neonatal pain assessment or NPA modelling).
•  In the methods, the technical approach is clear. However, relocating the subsections “A. Dataset Preparation and Processing” and “B. Implementation Details” from the results section to the methods would improve structural coherence. Quantitative findings such as grid search outcomes can remain in the results section.
•  In Table 1, formatting should be reviewed to ensure compliance with IEEE BHI guidelines, including placement (top or bottom of column) and font size requirements.
•  In Table 2, placement should follow IEEE BHI formatting requirements (i.e. top or bottom of a column).
•  In Figure 3, the figure placement should also be verified against IEEE BHI formatting rules.
•  In Table 2, the reported improvement in accuracy following the relabelling strategy would be strengthened by a statistical test to determine significance.
•  In Table 3, the performance advantage of the proposed dual-branch network over competing models should be supported by evidence of statistical significance to substantiate the claim.

---

### Official Review · Reviewer_pB4r · 2025-07-15
**Well structed but did they evaluate sex effect due to uneven numbers in genders?**

**Confidence:** 3
**Clarity Of Writing:** great
**Clinical Significance:** great
**Methodological Novelty:** great
**Overall Rating:** 7
**Final Rating:** 7

**Experiments And Results:**

great

**Questions For The Authors:**

Did you evaluate whether your model's performance differs between male and female neonates? Given the reported gender imbalance in the dataset, did you check that your model does not exhibit gender-based bias in pain assessment?

**Strengths:**

- The topic is interesting, and the gaps are clearly defined.
- The paper is well-structured.

**Summary Of The Paper:**

The paper proposes a Siamese network to reduce label uncertainty in neonatal pain assessment, achieving improved accuracy through dynamic label correction and attention-based analysis.

**Weaknesses:**

The paper does not analyze or control for potential gender-based bias in model performance, despite the dataset’s gender imbalance.

---

### Official Review · Reviewer_wBDJ · 2025-07-18
**Promising Framework for Neonatal Pain Classification, But Key Methodological Clarifications Needed**

**Confidence:** 3
**Clarity Of Writing:** great
**Clinical Significance:** good
**Methodological Novelty:** good
**Overall Rating:** 4
**Final Rating:** 7

**Experiments And Results:**

fair

**Questions For The Authors:**

There is considerable ambiguity regarding the data splitting and preprocessing process. The original dataset contains 613 video recordings, but the paper states that “each frame was independently scored by experienced hospital nurses” (page 4). The model was trained on individual image frames rather than on videos.
Several important clarifications are needed:
- Were all frames from a single subject assigned the same pain level label?
- If different frames from the same subject received different scores from experts, how was this handled during labeling?
- How were the image frames split into training and test sets? Specifically, do the training and test sets contain frames from the same subject?
- When manually introducing label noise, was the same biased label applied to all frames from a single subject, or were biases introduced independently per frame?
These questions are crucial to my assessment of the reliability of the experimental results. If these concerns regarding potential data leakage and inconsistent labeling are addressed satisfactorily, I will significantly raise my evaluation score.

Additionally, what percentage of the dataset consists of samples that are either unobstructed or show extreme facial positions? This information is important for assessing the robustness of the model in real-world, unconstrained settings.

**Strengths:**

The paper is well written, and all figures are clear.
It addresses an unconstrained environment, making the work more applicable to real-world scenarios.
The paper presents a comprehensive analysis to evaluate the effectiveness of label correction and the correction capability of the proposed method.

**Summary Of The Paper:**

This paper addresses the problem of automatically classifying the pain level of newborns in an unconstrained environment.
It utilizes a dual-branch self-correction mechanism to relabel potentially mislabeled image samples during training, thereby improving the robustness and accuracy of the model.
The results demonstrate effective label correction during training and improvements in prediction performance.

**Weaknesses:**

There is considerable ambiguity in how the data was split and preprocessed. I will elaborate on this concern in the “Questions for the Authors” section.

The sample images in Figure 4 are not representative enough to support the claim that the nose and mouth are critical regions for identifying severe pain. In the mild pain condition, these regions are occluded, making it unclear from the figure whether facial part occlusion or region-specific importance actually drives the model's decisions.

The paper claims novelty in using a heterogeneous branch compared to a homogeneous one, but no benchmark or comparison against a homogeneous version is provided—or at least, none is clearly indicated.

The length of the paper is 7 pages, which exceeds the BHI requirements (<=6pages).

The overall structure of the proposed mechanism is not clearly presented. While the paper explains the takeaways of each component well, it does not provide a clear picture of the complete architecture—particularly regarding how the Exponential Moving Average (EMA) mechanism relates to model updates. Additional explanation on how EMA updates specific model parameters would improve clarity. The current description is too general.

The noise used for evaluation is introduced manually, rather than derived from real-world data.

---

### Official Review · Reviewer_McKu · 2025-07-18
**Neonatal Pain Assessment under Label Uncertainty via Heterogeneous Siamese Learning**

**Confidence:** 3
**Clarity Of Writing:** good
**Clinical Significance:** great
**Methodological Novelty:** great
**Overall Rating:** 7
**Final Rating:** 7

**Experiments And Results:**

great

**Questions For The Authors:**

1. Are the authors willing to release the neonatal pain dataset used in this study?
2. Why was exactly 10% label noise chosen for simulation? Have the authors experimented with the model using the original annotations only (i.e., without artificial label noise)? How does performance compare in this setting?

**Strengths:**

1. The heterogeneous Siamese design increases diversity and helps better distinguish reliable vs. noisy samples, outperforming standard homogeneous dual networks.
2. Combines uncertainty weighting, rank regularization, and EMA-based pseudo-label refinement for label correction.
3. Grad-CAM analysis shows attention shifts from the eyes (mild pain) to the nose/mouth (severe pain), which aligns with clinical expectations and supports annotation feedback.
4. The proposed model achieves consistent performance improvements over Self-Cure and Erasing-Attention across five runs, with an average accuracy of 80.78%, outperforming all compared baselines.

**Summary Of The Paper:**

This paper addresses the challenging task of neonatal pain assessment (NPA) under uncontrolled clinical conditions, where facial images of neonates are often occluded or ambiguously labeled. The authors propose a dual-branch heterogeneous Siamese network that combines ResNet-50 and EfficientNet-V2 backbones to tackle label noise and annotation uncertainty. The method integrates uncertainty-guided rank regularization and mutual-supervision-based pseudo-label correction, enhanced by Exponential Moving Average (EMA) tracking. Grad-CAM visualizations provide interpretability by showing that the model attends to different facial regions depending on pain intensity. Experimental results demonstrate that the method achieves superior performance compared to recent state-of-the-art label correction techniques such as Self-Cure and Erasing-Attention.

**Weaknesses:**

1. The model is only evaluated on a private dataset; results on public pain datasets would improve generalizability claims.
2. Label noise is injected uniformly by shifting label values, which may not reflect real-world annotation complexities.

---

### Official Review · Reviewer_W2WW · 2025-07-20
**Bias correction technique**

**Confidence:** 4
**Clarity Of Writing:** good
**Clinical Significance:** good
**Methodological Novelty:** good
**Overall Rating:** 5
**Final Rating:** 7

**Experiments And Results:**

good

**Questions For The Authors:**

Please see the limitations.

**Strengths:**

1. Addresses an interesting problem of labeling bias correction.
2. Improves the performance by correcting the pain labels.

**Summary Of The Paper:**

This paper uses an uncertainty-aware dual-branch heterogeneous Siamese network to correct label biases for neonatal pain assessment (NPA).

**Weaknesses:**

1. However, the proposed method is not compared with other existing literature.
2. Pain is subjective. How do you find the real pain labels?
3. If you really have the actual pain labels, why not use them to train the model?
4. The performance is reported using only accuracy. A confusion matrix is better to demonstrate the holistic performance.